# Change in time spent visiting and experiences of green space following restrictions on movement during the COVID-19 pandemic: a nationally representative cross-sectional study of UK adults

Hannah Burnett  , Jonathan R Olsen  , Natalie Nicholls, Richard Mitchell

► Prepublication history and additional materials for this paper is available online. To view these files, please visit the journal online (http://dx.doi.org/10.1136/bmjopen-2020-044067).

MRC/CSO Social and Public Health Sciences Unit, University of Glasgow, Glasgow, UK

**Correspondence to**
Hannah Burnett;
h.burnett.1@research.gla.ac.uk

## ABSTRACT

**Objectives** Green space positively influences health and well-being; however, inequalities in use of green space are prevalent. Movement restrictions enforced due to the COVID-19 pandemic could have exacerbated existing inequalities regarding who visits green space. Therefore, this study aimed to explore how movement restrictions have changed the time spent visiting green space and experience of green space in the United Kingdom (UK) and how these differed by individual-level demographic characteristics.

**Design and outcome measures** A nationally representative cross-sectional survey administered through YouGov between 30 April and 1 May 2020. Data were collected on the time spent visiting green space and change in the experience of green space, including missing social interaction, increased physical activity and feeling greater mental health benefits in green space. Demographic information was collected on sex, age, ethnicity, social grade and dog ownership. Associations between specific outcome variables and predictors were assessed using logistic regression.

**Setting** UK, with population weights applied.

**Participants** 2252 adults aged 18 years and over.

**Results** Overall, 63% of respondents reported a decrease in time spent visiting green space following movement restrictions. Lower social grade respondents were less likely to visit green space before *and* after restrictions were enforced (OR: 0.35 (95% CI 0.24 to 0.51); OR: 0.77 (95% CI 0.63 to 0.95)). Female respondents were more likely than male respondents to agree that green space benefited their mental health more following restrictions (PP: 0.70 vs 0.59). Older (65+ years) respondents were less likely than middle-aged (25–64 years) respondents to have visited green space following the restrictions (OR: 0.79 (95% CI 0.63 to 0.98)).

**Conclusions** Inequalities in green space use were sustained, and possibly exacerbated, during movement restrictions. Our findings emphasise the importance of green spaces remaining open globally in any future 'lockdowns'/pandemics. Further investigation is required to determine how visit patterns and experiences change

### Strengths and limitations of this study

► Our data are currently the only existing data covering change in the time spent visiting green space and experiences of green space for the UK population following the movement restrictions enforced due to the COVID-19 pandemic.

► The sample is nationally representative of UK adults.

► Collecting data on *both* time spent visiting green space and change in experience of green space during the movement restrictions is a strength of this study, compared with other surveys exploring change in green space use during the COVID-19 pandemic, including the Google COVID-19 Community Mobility Reports and Natural England's People and Nature Survey.

► The data collected for this study were from a cross-sectional survey; therefore, it is not possible to demonstrate causality between change in green space use and experiences and the demographic data.

► Certain variable response categories were necessarily recategorised for analysis, which meant that some distinctiveness across groups was lost, particularly for the ethnicity and age variables.

through the different stages of the COVID-19 pandemic in the UK.

## INTRODUCTION

Evidence suggests that exposure to green space has a positive influence on health and well-being.[1][2] Green space use is associated with increased levels of social interaction and physical activity, and decreased levels of all-cause mortality, loneliness and stress.[3–5] Additionally, there is evidence that contact with green space may disproportionately benefit disadvantaged populations, reducing

BMJ

health inequalities and therefore weakening the effects of poverty, known as the 'equigenic' effect.[6 7]

The health impacts of contact with green spaces are quite well studied by both observational and experimental designs. There is rather less literature on the levels of, motivations for and barriers to visiting green spaces in the first place. Recent surveys by Natural England and Scottish Natural Heritage have found that 74% of the English population and 70% of the Scottish population visit green space 'frequently' (once a month or more often). In both countries, the top three reasons stated for frequent green space use were health/exercise, dog walking and to relax/unwind.[8 9] Further research exploring the motivations and reasons for visiting green space found that visitation frequency is affected by an individual's feelings of nature connectedness, as well as their childhood exposure to nature.[10–12] It is important to note that substantial inequalities in green space use have also been reported, with studies finding that females, older people and those from less advantaged socioeconomic positions (SEPs) are more likely to be infrequent users than their male, younger and higher SEP counterparts.[9 13–15]

In 2020, the United Kingdom (UK) experienced major disruption to everyday life due to the COVID-19 pandemic. COVID-19 is an infectious disease first identified in the city of Wuhan, China, in December 2019.[16] As a response to the COVID-19 outbreak, the UK announced a series of movement restrictions from 23 March 2020. These included rules designed to increase social distancing, meaning that people could only leave their households to make 'essential trips' for food, medication and exercise.[17] Recent studies have highlighted the negative effects of COVID-19 on the UK population's mental health and well-being, which are likely to be profound and long lasting.[18] Research exploring the wider health effects of COVID-19 suggest that the negative indirect effects are being borne disproportionately by people who have fewer resources and poorer health.[19] If natural environments usually act to mitigate the connections between adversity and poor health,[6 7] it is important to assess the extent to which lockdown affected both use and experience of such environments. Future lockdowns and movement restrictions are highly likely as second, third and perhaps fourth waves of the pandemic take place around the world.

The aim of this research was to explore changes in the time spent visiting green space and experience of visits to green space among the UK population following movement restrictions being enforced. The sociodemographic characteristics focused on were those identified by the literature as being most consistent markers of inequality in use. The research questions were:

1. How did time spent visiting green space change following movement restrictions, compared with before, and for whom?
2. How did experiences of visits to green space change following the restrictions, and did any change vary by sex, age, ethnicity, social grade and dog ownership? Specifically, (A) did respondents feel that green space

benefited their mental health more since the movement restrictions were enforced than before? (B) Did respondents miss social interaction in green space more following movement restrictions? (C) Had physical activity increased following movement restrictions?

## METHODS

### Survey design and sample

An online cross-sectional survey was administered by YouGov between 30 April and 1 May 2020.[20] Questions were answered by a sample of 2252 adults from the UK (aged 18 years and over). The sample was drawn from a panel of over 800 000 individuals who specifically opted in to participate in online research activities. Sample members were randomly selected from the panel and sent an email providing a survey link. Table 1 shows the themes and specific survey instruments analysed in this paper. Only the respondents that visited green space following movement restrictions were asked the questions regarding change in experience of green space. Demographic information about participants was also collected (including sex, age and social grade, which was classified by occupation).[21 22] Although the sample was reasonably large, small numbers in variable response categories necessitated some category mergers (table 1 and online supplemental table 1).

The survey covered adults from across the UK, with respondents from England (n=1875), Scotland (n=209), Wales (n=107) and Northern Ireland (n=61). Weightings were applied to render the sample representative of UK adults (detailed below). When the survey was distributed, the same movement restrictions were implemented across the UK. These included only leaving home for limited purposes, such as medical needs, shopping for basic necessities (food and medicine) and exercising once a day alone/with members of your household.[23] Since then, the individual parliaments/assemblies representing these countries have imposed different COVID-19 policies.

### Demographic variables

Individual-level demographic and socioeconomic characteristics were captured from the survey, as follows: sex (male or female); age group (18–24 years, 25–64 years and 65+ years); ethnicity (white, black, Asian and minority ethnic (BAME)); dog ownership (yes or no); and social grade (ABC1 and C2DE), derived by YouGov from combined occupational social grade categories. ABC1 was the higher social grade and included non-manual workers, for example, senior managers and owners of small establishments. C2DE was the lower social grade and included all manual workers, for example, shop assistants and labourers.[24] Hereafter, social grade will be described as higher and lower social grade.

### Patient and public involvement

There was no direct patient or public involvement in this study.

**Table 1** Survey themes and specific items analysed, including variables recategorised

| Themes | Question/statement | Potential responses | Recategorised |
|---|---|---|---|
| Change in the amount of time spent visiting green space | 'Did you EVER visit a green space in the year before the movement restrictions were enforced in the UK?'. | Yes, I did | – |
| | | No, I didn't | - |
| | | Don't know/can't recall | Don't know/can't recall excluded (n=80, 3.5%) |
| | 'Please think about your behaviour since the UK enforced a "lockdown" to restrict movement, as a result of the current Coronavirus (COVID-19) outbreak (ie, since 23rd March 2020). Have you visited a green space since the movement restrictions have been enforced in the UK?'. | Yes, I have | – |
| | | No, I haven't | - |
| | | Don't know/can't recall | Don't know/can't recall excluded (n=43, 1.9%) |
| | 'How much, if at all, has the amount of time that you have spent visiting green spaces changed since the "lockdown" movement restrictions began (ie, 23rd March 2020) compared to before?'. | Increased a lot | 'Increased' (increased a lot and increased a little) |
| | | Increased a little | 'Increased' (increased a lot and increased a little) |
| | | No difference | 'Same' (no difference) |
| | | Decreased a little | 'Decreased' (decreased a lot and decreased a little) |
| | | Decreased a lot | 'Decreased' (decreased a lot and decreased a little) |
| | | Don't know | Don't know excluded (n=74, 3.3%) |
| Experience change (If respondent had visited a green space since the movement restrictions were enforced) | 'I feel that being in green spaces benefits my mental health more now, than before the movement restrictions were in place'. | Strongly agree | 'Agree' (strongly agree and slightly agree) |
| | | Slightly agree | 'Agree' (strongly agree and slightly agree) |
| | | Neither agree nor disagree | 'Neither' (neither agree nor disagree) |
| | | Slightly disagree | 'Disagree' (slightly disagree and strongly disagree) |
| | | Strongly disagree | 'Disagree' (slightly disagree and strongly disagree) |
| | | Don't know/can't recall | Don't know/can't recall excluded (n=7, 0.6%) |
| | 'I have missed seeing and/or talking with people in green spaces since the movement restrictions were introduced'. | Strongly agree | 'Agree' (strongly agree and slightly agree) |
| | | Slightly agree | 'Agree' (strongly agree and slightly agree) |
| | | Neither agree nor disagree | 'Neither' (Nneither agree nor disagree) |
| | | Slightly disagree | 'Disagree' (slightly disagree and strongly disagree) |
| | | Strongly disagree | 'Disagree' (slightly disagree and strongly disagree) |
| | | Don't know/can't recall | Don't know/can't recall excluded (n=8, 0.8%) |

Continued

**Table 1** Continued

| Themes | Question/statement | Potential responses | Recategorised |
|---|---|---|---|
| | 'I do more physical activity in green spaces now, than I did before the movement restrictions were introduced'. | Strongly agree | 'Agree' (strongly agree and slightly agree) |
| | | Slightly agree | 'Agree' (strongly agree and slightly agree) |
| | | Neither agree nor disagree | 'Neither' (Nneither agree nor disagree) |
| | | Slightly disagree | ''Disagree'' (slightly disagree and strongly disagree) |
| | | Strongly disagree | ''Disagree'' (slightly disagree and strongly disagree) |
| | | Don't know/can't recall | Don't know/can't recall excluded (n=9, 0.8%) |

Don't know/can't recall responses were all excluded from analysis (weighted counts and proportions reported above).

## Analyses

### Descriptive statistics

The count and proportion of respondents who had: visited green space before and after movement restrictions were enforced; increased or decreased visitation; and agreed or disagreed with the three change in experience statements (table 1) are presented. These were also explored by sex, age, ethnicity, social grade and dog ownership.

### Statistical analysis

Binary logistic regression analyses were conducted to assess the association between the individual predictor variables and the following two outcomes:
1. Having visited green space in the year before movement restrictions were enforced.
2. Having visited green space since movement restrictions were enforced, with this analysis being restricted to those who had reported visiting green space before movement restrictions.

The associations between the individual predictor variables and each of the following outcomes were also assessed using multinomial logistic regression or, if appropriate, ordinal logistic regression, with results presented as predicted probabilities (PPs):
1. Change in green space visitation following movement restrictions.
2. Levels of agreement that green space benefits their mental health more now (since movement restrictions were enforced compared with before).
3. Levels of agreement that they miss seeing/talking to people in green space now (since movement restrictions were enforced compared with before).
4. Levels of agreement that they do more physical activity in green space now (since movement restrictions were enforced compared with before).

The results of the binary logistic regression analyses were expressed as odds ratios (ORs) with 95% confidence intervals (CIs). Other results are presented as predicted outcome group probabilities for each variable. PPs can be interpreted as an indicator of likelihood, so that the closer the value is to 1.0, the greater the likelihood. We opted to present these results as PPs as they better illustrate the size of the association between the predictor variable and response category and the difference in this *between* variables. For example, the likelihood of men either decreasing, maintaining or increasing their time in green space can be more easily compared with the likelihood of those in the higher social grade category decreasing, maintaining or increasing their time in green space using PP than with an OR or risk ratio (RR). This is because an OR or RR is expressed relative to the reference category. For those who prefer an RR, these are provided in the supplementary material (online supplemental tables 2–5).

Univariate models for each predictor were conducted first, followed by a fully adjusted model containing all predictors for each outcome. Weightings were calculated by YouGov, with the final data weighted to match the national profile of all adults aged 18 years and over and applied during analyses to render the sample representative of UK adults.[22] All analyses were conducted in R V.3.5.1,[25] and the *brant* package,[26] was used to check that the proportionality of odds assumption for ordinal logistic regression was not violated.[27] A full R script is available on GitHub.[28]

## RESULTS

### Descriptive statistics

Ninety-three per cent of all respondents had visited green space in the year before movement restrictions were enforced. In contrast, 53% of respondents reported visiting green space following movement restrictions. Sixty-three per cent of respondents reported that the

amount of time they spent in green space had decreased since movement restrictions compared with before, with 15% reporting an increase and 22% reporting no difference in the time spent visiting green space (online supplemental table 6).

A greater proportion of respondents agreed (65%) than disagreed (10%) or neither agreed nor disagreed (25%) that green space benefited their mental health more following movement restrictions being enforced compared with before. More respondents agreed (54%) that they missed social interaction in green space more since movement restrictions than disagreed (19%) or neither agreed nor disagreed (27%). Thirty-nine per cent of respondents disagreed that they had increased physical activity in green space since movement restrictions, compared with 29% who agreed and 32% that neither agreed nor disagreed (online supplemental table 7).

### Change in visitation time
#### Visiting green space before movement restrictions
In the adjusted logistic regression model for visiting green space in the year before movement restrictions were enforced (table 2), only two variables had significant associations (p<0.05). Lower social grade respondents (OR: 0.35 (95% CI 0.24 to 0.51)) and BAME (OR: 0.43 (95% CI 0.23 to 0.80)) respondents had lower odds of visiting green space before movement restrictions, compared with higher social grade and white respondents.

#### Visiting green space following movement restrictions
Respondents in the lower social grade group were less likely to have visited green space than respondents in the higher social grade group (OR: 0.77 (95% CI 0.63 to 0.95)) (table 2). Older respondents (65+ years) were also less likely than middle-aged respondents (25–64 years) to have visited green space following the movement restrictions (OR: 0.79 (95% CI 0.63 to 0.98)). Respondents who owned a dog/s were more likely than respondents who did not to have visited green space after movement restrictions were enforced (OR: 1.42 (95% CI 1.14 to 1.78)).

#### Change in time spent visiting green space
Following lockdown, changes in the time spent visiting green space was found to differ by demographic group. The likelihood of spending more time in green space was found to differ by social grade. Lower social grade respondents were less likely to report spending more time in green space following movement restrictions than higher social grade respondents (PP: 0.09 vs 0.16) (table 3). Age was also found to be associated with change in time spent in green space. Older respondents (aged 65+ years) were the least likely to report increased green space visits (PP: 0.09 (65+ years) compared with younger groups (PP: 0.14 (25–64 years) and 0.21 (18–24 years))). Females were more likely to report decreased green space visits compared with males (PP: 0.67 vs 0.62), as well as being less likely to report no change in visit time (PP: 0.20 vs 0.25) (table 3). Finally, respondents without a dog/s were slightly more

**Table 2** Adjusted binary logistic regression models predicting green space visit before and after the movement restrictions were enforced; *p<0.05.*

| | Visited green space in the year before restrictions (yes) | | Visited green space after restrictions (yes and visited green space before restrictions) | |
|---|---|---|---|---|
| | Adjusted | | Adjusted | |
| | OR (95% CI) | P value | OR (95% CI) | P value |
| **Sex** | | | | |
| Male (ref) | | | | |
| Female | 1.35 (0.94 to 1.95) | 0.109 | 0.88 (0.72 to 1.07) | 0.205 |
| Social grade | | | | |
| ABC1 (ref) | | | | |
| C2DE | 0.35 (0.24 to 0.51) | *<0.001* | 0.77 (0.63 to 0.95) | *0.013* |
| Age (years) | | | | |
| 18–24 | 2.92 (1.00 to 8.57) | 0.051 | 0.81 (0.54 to 1.21) | 0.297 |
| 25–64 (ref) | | | | |
| 65+ | 1.22 (0.80 to 1.85) | 0.353 | 0.79 (0.63 to 0.98) | *0.035* |
| Ethnicity | | | | |
| White (ref) | | | | |
| BAME | 0.43 (0.23 to 0.80) | *0.007* | 1.06 (0.67 to 1.68) | 0.799 |
| Dog ownership | | | | |
| No (ref) | | | | |
| Yes | 1.29 (0.83 to 2.00) | 0.26 | 1.42 (1.14 to 1.78) | *0.002* |

The italics are when the p-value is <0.05.
BAME, black, Asian and minority ethnic.

likely to report decreased green space visitations compared with dog owners (PP: 0.66 vs 0.62) and less likely to have sustained their frequency of visitation (PP: 0.21 vs 0.28).

### Change in visit experience
#### Mental health benefits
Females were more likely to agree than males that being in green space benefited their mental health more following movement restrictions than before (PP: 0.70 vs 0.59). Higher social grade respondents were more likely to agree than lower social grade respondents (PP: 0.68 vs 0.59). Younger respondents were more likely to disagree that being in green space benefited their mental health more following movement restrictions than before, while older respondents were less likely to disagree (PP: 0.25 (18–24) vs 0.06 (65+ years) and 0.10 (25–64 years)) (table 3).

#### Missed social interaction
Female respondents were more likely to agree that they missed seeing and talking with other people in green

**Table 3** Multinomial logistic regression models: predicted probabilities (likelihoods) of being in each outcome group for change in time visiting green space and levels of agreement that green space benefits their mental health more now (since movement restrictions were enforced compared with before); $p<0.05$.

| | Change in time spent visiting green space | | | | Using green spaces benefits my mental health more now | | | |
|---|---|---|---|---|---|---|---|---|
| | Decreased | Same | Increased | P value | Agree | Neither | Disagree | P value |
| Sex | | | | | | | | |
| Male | 0.62 | 0.25 | 0.13 | *0.041* | 0.59 | 0.31 | 0.1 | *0.004* |
| Female | 0.67 | 0.2 | 0.12 | | 0.7 | 0.22 | 0.09 | |
| Social grade | | | | | | | | |
| ABC1 | 0.65 | 0.19 | 0.16 | *<0.001* | 0.68 | 0.24 | 0.09 | *0.048* |
| C2DE | 0.64 | 0.27 | 0.09 | | 0.59 | 0.31 | 0.1 | |
| Age (years) | | | | | | | | |
| 18–24 | 0.57 | 0.22 | 0.21 | *0.004* | 0.55 | 0.2 | 0.25 | *<0.001* |
| 25–64 | 0.63 | 0.22 | 0.14 | | 0.68 | 0.22 | 0.1 | |
| 65+ | 0.69 | 0.22 | 0.09 | | 0.55 | 0.38 | 0.06 | |
| Ethnicity | | | | | | | | |
| White | 0.22 | 0.65 | 0.12 | 0.167 | 0.64 | 0.26 | 0.1 | 0.063 |
| BAME | 0.3 | 0.56 | 0.14 | | 0.73 | 0.25 | 0.02 | |
| Dog ownership | | | | | | | | |
| No | 0.66 | 0.21 | 0.13 | *0.003* | 0.67 | 0.24 | 0.09 | 0.117 |
| Yes | 0.62 | 0.28 | 0.1 | | 0.59 | 0.3 | 0.11 | |

The p values presented reflect the significance of each factor in the model.
The italics are when the p-value is <0.05.
BAME, black, Asian and minority ethnic.

space since movement restrictions compared with before than male respondents (PP: 0.58 vs 0.45) (table 4).

### Increased physical activity

Older respondents were less likely to agree that they had increased physical activity following movement restrictions (PP:0.18 (65+) vs 0.29 (25–64) and 0.44 (18–24)). Respondents who owned a dog/s were less likely than respondents without a dog/s to agree (PP: 0.17 vs 0.31) (table 4).

### DISCUSSION

Our findings suggest that inequalities in use of green space between demographic groups were sustained following movement restrictions, with lower social grade individuals less likely than higher social grade individuals to have visited green space before and since the movement restrictions were introduced. Other existing inequalities in use were possibly exacerbated in the month after movement restrictions were enforced, with females being more likely to report a decrease in visits following movement restrictions.

The proportion of respondents who visited green space before and following movement restrictions decreased from 93% to 53%. This was consistent with Natural England's findings from April 2020 where 49% of English adults reported green space visits in the previous

2 weeks.[29] Natural England conducted an online panel survey in April 2020 (n=2083); the survey covered only the English population, collecting information on the frequency of time in green and natural spaces in the 12 months, and then 2 weeks, prior to the survey.[30] The survey did not directly ask respondents whether they felt their experiences within green space had changed following movement restrictions. A similar study focused on change in time spent visiting parks using the Google COVID-19 Community Mobility Reports covering 620 counties across the USA. They found a lower percentage decrease in park visits compared with our findings, reporting a 17%–35% decrease in visits between 15 March and 9 May 2020.[31] This difference may be explained by the focus on parks alone rather than different types of green space. However, additional research exploring the Google COVID-19 Community Mobility Reports found that from 16 February to 29 March 2020, park use decreased by 90% in Catalonia, 7% in Oslo and 79% in New York County. In Stockholm, park use increased by 24% in the same timeframe.[32] This could be explained by Sweden having a less stringent approach to movement restrictions and instead relying on 'self-responsibility' to prevent the population having to restrict movement and stay at home.[33] The Swedish population may also be more culturally attuned to seeking time in nature to combat stress.

**Table 4** Multinomial logistic regression models: predicted probabilities (likelihoods) of each levels of agreement that respondents missed seeing/talking to people in green space more since movement restrictions were enforced compared with before, and levels of agreement that respondents do more physical activity in green space following the movement restrictions; *p<0.05.*

| | Miss social interaction in green spaces now | | | | Do more physical activity in green spaces now | | | |
|---|---|---|---|---|---|---|---|---|
| | Agree | Neither | Disagree | P value | Agree | Neither | Disagree | P value |
| Sex | | | | | | | | |
| Male | 0.45 | 0.3 | 0.25 | *<0.001* | 0.24 | 0.37 | 0.4 | 0.096 |
| Female | 0.58 | 0.26 | 0.16 | | 0.27 | 0.29 | 0.43 | |
| Social grade | | | | | | | | |
| ABC1 | 0.53 | 0.27 | 0.2 | 0.256 | 0.27 | 0.31 | 0.42 | 0.307 |
| C2DE | 0.48 | 0.31 | 0.21 | | 0.23 | 0.36 | 0.41 | |
| Age | | | | | | | | |
| 18–24 | 0.59 | 0.23 | 0.18 | 0.672 | 0.44 | 0.23 | 0.33 | *0.002* |
| 25–64 | 0.51 | 0.28 | 0.21 | | 0.29 | 0.31 | 0.4 | |
| 65+ | 0.53 | 0.3 | 0.18 | | 0.18 | 0.38 | 0.45 | |
| Ethnicity | | | | | | | | |
| White | 0.52 | 0.28 | 0.2 | 0.802 | 0.25 | 0.32 | 0.42 | 0.063 |
| BAME | 0.52 | 0.31 | 0.16 | | 0.33 | 0.44 | 0.22 | |
| Dog ownership | | | | | | | | |
| No | 0.5 | 0.3 | 0.2 | 0.295 | 0.31 | 0.3 | 0.39 | *<0.001* |
| Yes | 0.55 | 0.25 | 0.2 | | 0.17 | 0.38 | 0.46 | |

The p values presented reflect the significance of each factor in the model.
The italics are when the p-value is <0.05.
BAME, black, Asian and minority ethnic.

Females and older individuals were more likely to have reported a decrease in visits to green space following movement restrictions being enforced. These findings corroborate with existing data collected before movement restrictions.[9 13 14] Boyd *et al*[9] analysed the Natural England 'Monitor of Engagement with the Natural Environment' survey (n=63 890) with a focus on infrequent use of green space. They found that females and older adults in England were more likely to be infrequent visitors.[9] Cohen *et al*[13] explored physical activity levels in Los Angeles' parks (n=1318). They reported that age and sex were predictors of park use, with <5% of park users being over 60 years old, and males using parks more often than females (62% vs 38%).[13]

Our findings suggested that existing patterns of gender inequality in use were sustained and potentially exacerbated due to fears arising from the COVID-19 pandemic. For example, an Ipsos MORI survey was conducted in April 2020, collecting data on gender differences in British attitudes towards COVID-19. They found that women were more likely to report following government rules to avoid leaving their home (78%) than men (68%) and felt more uncomfortable about returning to 'normal' (n=1000).[34 35]

Previous studies have found that females feel more vulnerable than males in green spaces, especially without company.[14 15 36–39] There are few quantitative studies focusing on sex and the importance of social interaction as a motivation behind green space use. We found that females were more likely to agree that they missed seeing/talking with others in green space than males. This could be explained by the nature of the movement restrictions at this time, particularly the inability to socially interact with individuals outside your household in green space.

Given older age is probably the most important risk factor for an adverse outcome from COVID-19,[40 41] it was not surprising that older individuals were less likely to report an increase in green space visits and to agree that they were doing more physical activity following movement restrictions being enforced. These findings corroborate with the majority of research exploring older age and green space use before the movement restrictions.[9 13 42] Older people in the UK have been found to be at particular risk of social isolation during social distancing, being less likely to use online communications and more likely to live alone than younger individuals.[19] Our findings also show that older people are less likely to have used green space during lockdown, further emphasising this point. The significant decrease in green space visits following movement restrictions for older respondents could be explained by the government advice for over 70s and those with an underlying health condition to shield, minimise interaction and stay at home for around 12 weeks from 21 March 2020.[43]

Lower social grade respondents reported little or no change in visitation to green space, with time spent visiting green space remaining low both before and after movement restrictions were enforced. This is supported by previous studies exploring the association between socio-economic position (SEP) and green space use.[9 15 42] One possible explanation is a lack of interest in visiting green space reported by lower SEP individuals.[9] Additionally, the social grade variable was categorised by occupation, and it was likely that individuals categorised as lower social grade were working in manual or service occupations.[24] They may have continued at their usual workplace/working hours during the COVID-19 pandemic. Findings from the Office for National Statistics support this explanation, with 30.5% of employees in the bottom three income deciles (monthly earnings of up to £1450) considered as key workers in March/April 2020, compared with 26.4% in the top three income deciles (monthly earnings of up to £3250).[44] Similarly, existing international research on public space use during COVID-19 restrictions worldwide states that skilled workers in the knowledge economy have shifted easily into online work from home and can therefore make more use of green spaces during the movement restrictions.[45]

The results for dog owners suggest that the movement restrictions have had an overall negative impact on their experiences within green space. Although dog owners were more likely to have visited green space following the movement restrictions than those that did not own a dog, they were less likely to agree that they had increased physical activity and more likely to have decreased visitations following the restrictions. These results differ from research undertaken in Canada that reported findings that dog ownership was associated with more outdoor play and less indoor play in Canadian youth at the start of the pandemic.[46] However, the focus on youth rather than adults may explain this difference in findings. Instead, the difference found in visits after restrictions were enforced may be due to dog owners having to walk their dog/s in green spaces *despite* the pandemic restrictions. In comparison between our results and those of other studies, we are mindful of the difference between number of visits and time spent in green spaces as measures of 'use'. It would be entirely possible to increase one at the expense of the other, and our data were not well suited to unpacking the relationship between them.

### Strengths and limitations

Our study has several strengths; to our knowledge, currently this is the only data covering UK population change in use and experiences within green space at the start of the movement restrictions from 23 March 2020. This means that the data could provide timely information to local and national governments across the UK. It is important to understand the effects of the initial movement restrictions to generate policy recommendations for any future movement restrictions. The rapid collection of data, just over a month after the movement restrictions were introduced, reduces risk of recall bias. The sample is nationally representative of UK adults, with weightings calculated by YouGov and applied to all analyses, reducing risk of selection bias.[22]

A further strength of our study is providing quantitative data on the importance of social interaction as a driver for using green space, which is under-reported in the literature. We analysed social grade at an individual level, which provides greater detail and accuracy of the respondent's SEP than at neighbourhood level.

There are some caveats. The survey design was cross-sectional, and we are unable to demonstrate causality between change in green space use and experiences and the demographic data. Additionally, the data on use and experiences are self-reported and therefore subjective. Certain variable response categories had to be recategorised in order to analyse them. This is particularly evident in the ethnicity and age variables, where some distinctiveness across groups was lost. The variable best capturing change in behaviour measured time spent in green space, but this made it difficult to understand whether respondents traded off time and numbers of visits.

We encourage future research to explore the reasons *why* many patterns and changes in use of green space since movement restrictions were enforced have emerged. It would also be interesting to explore the change in patterns of use before and following movement restrictions for those that did not visit before the movement restrictions but did visit green space following restrictions.

### CONCLUSION

Our study provides novel evidence to suggest that green space use and experiences were profoundly affected during the first month of movement restrictions in the UK, which were part of the response to COVID-19. Our findings suggested that inequalities in use were sustained, with lower social grade individuals less likely to visit green space than higher social grade individuals before and following movement restrictions. It is possible that these inequalities were exacerbated, as females and older individuals were the groups most likely to have decreased visits following movement restrictions. Although these findings reflect the UK population's experience during the movement restrictions, they could be compared with countries such as Norway, USA and Spain, where green space visits also decreased. We believe that these findings emphasise the need for green spaces to remain open in any future 'lockdowns' and for governments to actively encourage individuals to use these spaces to support their mental and physical health during subsequent waves of the pandemic. Further investigation is required to support these findings and how they may change through the stages of the COVID-19 pandemic and the relaxing, and potential reintroduction, of movement restrictions.

**Contributors** HB, JRO and RM designed the study and survey; HB conducted data analysis, which was reviewed by NN; HB wrote the first draft of the paper;

all authors contributed to interpreting the results and revising the draft. The corresponding author attests that all listed authors meet authorship criteria and that no others meeting the criteria have been omitted.

**Funding** All authors are part of the Places and Health Programme at the MRC/CSO (Medical Research Council/Chief Scientist Office) Social and Public Health Sciences Unit, University of Glasgow, supported by the Medical Research Council (MC_UU_12017/10) and the Chief Scientist Office (SPHSU10). HB is also funded by a Medical Research Council and University of Glasgow College of Medical, Veterinary and Life Sciences PhD studentship (MC_ST_U18004).

**Competing interests** None declared.

**Patient consent for publication** Not required.

**Provenance and peer review** Not commissioned; externally peer reviewed.

**Data availability statement** Data are available in a public, open access repository. We have made our research dataset publicly available (Datacite DOI: 10.5525/gla.researchdata.1038 - embargoed until 31 October 2023).

**ORCID iDs**
Hannah Burnett http://orcid.org/0000-0001-5710-3180
Jonathan R Olsen http://orcid.org/0000-0002-5356-8615

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
