## [Reviewer comments · BMJ Open]

ARTICLE DETAILS

TITLE (PROVISIONAL)	Change in time spent visiting and experiences of green space following restrictions on movement during the COVID-19 pandemic: A nationally representative cross-sectional study of UK adults.
AUTHORS	Burnett, Hannah; Olsen, Jonathan; Nicholls, Natalie; Mitchell, Richard

VERSION 1 – REVIEW

REVIEWER	Judith Vonk University Medical Center Groningen, Groningen, The Netherlands
REVIEW RETURNED	16-Sep-2020

GENERAL COMMENTS	The authors describe a questionnaire study in which they investigated the change in frequency and experience of green space visits before and during the COVID-lockdown in the UK. The background of the question is to investigate if the lockdown has exacerbated existing inequalities in green space visits. The results show an overall decrease in green space visits and the existing inequalities were sustained, and possibly exacerbated. I have the following comments: 1. My main comment is about the relevance of the study. Why is it important to know this? Who will benefit from this knowledge and what can be done with the results and how? I agree that it is just nice to know this and therefore I recommend that the paper should be rewritten as a short report (leaving out the interactions and performing multinomial regressions and combine question 1 and 2 (see below)).2. Which interactions were investigated and why did the authors expect these interactions? More information should be provided. I also do not like figure 1 since this has nothing to do with the COVID restrictions but only gives some information about before the COVID crisis (also do not use bars but give a point estimate with a CI here). Finally, present the results of all investigated interactions.3. It is unclear if the questions on experience of green space were asked to all respondents or only to those who indeed visited green spaces (during lockdown).4. How exactly was the weighting done5. To be able to say something about a real change in the prevalence of visiting green space the authors should have used the pairedness of the data. If they compare question 1 and 2 (visit
--

	green space before and during the lockdown) they could have made specific combinations (yes/yes, yes/no, no/yes, no/no). Especially the yes/no group is interesting to see if there is exacerbated inequality (e.g. are there more C2DE subjects in this group compared to the yes/yes and also is this different from the ABC1 group). But also the no/yes group could be interesting to investigate further. 6. Instead of doing binary logistic regression on decrease and increase separately the authors could also have done multinomial logistic regression in which they used the 3 categories (increased, the same, decreased) as their outcome (using 'the same' as reference category). This also applies to the agree/disagree variables. 7. In the supplementary tables the numbers do not always add up to the total. For example in S1 1659 white british and 86 any other white background makes 1659+86=1745 and not 1747 as stated in the table. Also in table S2 158 did not visit green space before restriction of which 119 did not own a dog and 40 owned a dog, but 119+40=159 and not 158. I did not check everything but I urge the authors to check all the numbers.
--	--

REVIEWER	Zander Venter Norwegian Institute for Nature Research
REVIEW RETURNED	28-Sep-2020

GENERAL COMMENTS	The authors present the results of a cross-sectional survey in the UK exploring the effects of COVID-19 lockdowns on the visitation to and experience of green spaces. They find an overall decline in self-reported visitation to green spaces with demographic differences that indicate a possible exacerbation of inequality in use of green space. The manuscript is well written and will be of interest to readers of BMJ Open. The findings and conclusions drawn are supported by the methods and results presented. I have only minor comments below. P4L35: I find myself wanting to know more detail about the social distancing measures here. It is important for an international audience who might have questions about what constituted “restrictions on movement” in the UK. For instance: were there any curfews in place? What was the scope of “exercise” – i.e. did it include walking pets? P5L41: The wording of the question in the survey does not necessarily imply frequency of visitation. The words “...time that you have spent visiting green spaces...” may be interpreted as number of visits or as time spent during each visit. In this way, two respondents may have answered “Increased” and one may have increased the frequency of green space visits while the other visited the same number of times as before the ‘lockdown’ but spent a longer duration of time in the green space. I would argue that the use of “frequency” in the title and throughout the manuscript is a little misleading given that the survey was not this specific. P6L29: It is not clear how what the significance of the dog ownership classification is. The results are reported but not discussed in the Discussion section. What are the implications of the results related to dog ownership? P10L35: Discussion: this section is well written, however lacks reference to broader literature on the topic which decreases the
---

	relevance for an international audience. I advise including similar literature from other countries in Europe and further afield. P17 Figure 1: Please remind the reader what “ABC1” and “C2DE” mean here.
--	--

VERSION 1 – AUTHOR RESPONSE

Reviewer: 1

Comments for the authors

The authors describe a questionnaire study in which they investigated the change in frequency and experience of green space visits before and during the COVID-lockdown in the UK. The background of the question is to investigate if the lockdown has exacerbated existing inequalities in green space visits. The results show an overall decrease in green space visits and the existing inequalities were sustained, and possibly exacerbated. I have the following comments:

1. My main comment is about the relevance of the study. Why is it important to know this? Who will benefit from this knowledge and what can be done with the results and how? I agree that it is just nice to know this and therefore I recommend that the paper should be rewritten as a short report (leaving out the interactions and performing multinomial regressions and combine question 1 and 2 (see below)).

Response: The early restrictions on movement in the UK were a unique situation that could have long-lasting impacts on health, with unprecedented restrictions in the use of public spaces, including green spaces, worldwide. The UK Government recognised the value of access to urban green space and required councils to keep parks open during the lockdown, emphasising the importance placed on these spaces. This guidance differed globally, with countries such as Spain and Italy closing green spaces.[1,2] Therefore, this study provides evidence from a country where the majority of public green spaces remained open of the influence that green space can have on mental health, social interaction and physical activity during the pandemic.

It is important to understand the effect of the lockdown on green space use and for different demographic groups in order to generate policy recommendations for governments if there are future lockdowns (as is highly likely given the current evidence on loss of immunity to COVID-19) or pandemics. Across Europe governments are currently imposing new restrictions on movement, therefore our results have significance for aiding policy decisions, including park opening decisions, and potential impacts of these restrictions on population level mental health outcomes. Additionally, the results of this paper could be utilised in both academia and policy to stimulate further research into how the pandemic and related restrictions have and will impact on green spaces and populations both in the UK and elsewhere in the world. The results in this study could be compared to any future research conducted on green space use during the different stages of the COVID-19 pandemic. We have added the following sentences to emphasise the importance of this research:

“It is important to understand the effects of the initial restrictions on movement to generate policy recommendations for any future movement restrictions.” (Page 17, lines 519-521)

“Recent studies have highlighted the negative effects of COVID-19 on the UK population’s mental health and wellbeing, which are likely to be profound and long-lasting.[18] Research exploring the wider health effects of COVID-19 suggest that the negative indirect effects are being borne disproportionately by people who have fewer resources and poorer health.[19] If natural environments usually act to mitigate the connections between adversity and poor health,[6,7] it is important to assess the extent to which lockdown affected both use and experience of such environments. Future lockdowns and movement restrictions are highly likely as second, third and perhaps fourth waves of the pandemic take place around the world.” (Page 4, lines 119-127)

The design of our study is also important, adding to the global understanding of the unintended consequences of restrictions on movement on population health. For example, our focus on missing social interaction in green spaces adds further evidence to suggest that negative indirect effects of the pandemic and lockdown are borne disproportionately by people who already have fewer resources and poorer health. Older people in the UK have been found to be at particular risk of social isolation during social distancing, being less likely to use online communications and more likely to live alone than younger age groups.[3] Our findings also show that older people are less likely to have used green space during lockdown, further emphasising this point.

References:

- 1 ***Ren X. Pandemic and lockdown: a territorial approach to COVID-19 in China, Italy and the United States. Eurasian Geogr Econ 2020;:1–12. doi:10.1080/15387216.2020.1762103***
- 2 ***Agencia Estatal Boletín Oficial del Estado. Royal Decree 462/2020, of March 14, which declares the state of alarm for the management of the health crisis situation caused by COVID-19. BOE. 2020.<https://www.boe.es/buscar/doc.php?id=BOE-A-2020-3692>***
- 3 ***Douglas M, Katikireddi SV, Taulbut M, et al. Mitigating the wider health effects of covid-19 pandemic response. BMJ 2020;369. doi:10.1136/bmj.m1557***

2. Which interactions were investigated and why did the authors expect these interactions? More information should be provided. I also do not like figure 1 since this has nothing to do with the COVID restrictions but only gives some information about before the COVID crisis (also do not use bars but give a point estimate with a CI here). Finally, present the results of all investigated interactions.

Response: Interactions were investigated as an exploratory analysis between the following demographic variables: sex, social grade, age, and ethnicity. This was because these variables

have been identified as important moderators of green space use in existing literature. We have made this clearer by amending the following sentence:

“Given existing literature,[9,13,29,30] we expected some interactions between the demographic variables sex, age, social grade, and ethnicity, in their relationships with change in visit time and experience of green space following movement restrictions. We therefore explored interactions between each of these demographic variables for every model, assessing their significance via Wald tests, and then producing predicted probabilities to aid interpretation of the significant interactions.” (Page 8, lines 247-253)

The interactions were investigated for all of the models and demographic variables listed above, therefore we have not presented all of the results. We have now added the significant interaction results to the supplementary materials (Supplementary Table 4 and 5).

Following the suggestion of the reviewer, we have changed Figure 1 to now present the results of the interactions between ethnicity and age for the change in time spent in green space following movement restrictions.

3. It is unclear if the questions on experience of green space were asked to all respondents or only to those who indeed visited green spaces (during lockdown).

Response: The questions on change in experience of green space were asked to only those respondents that had visited green space during the restrictions on movement. We have added the following text to clarify this:

“Only the respondents that visited green space following movement restrictions were asked the questions regarding change in experience of green space.” (Page 4, lines 158-160)

4. How exactly was the weighting done

Response: Once the survey was complete, weightings were applied to the final sample to the national profile of all adults aged 18+ (including people without internet access). This process was undertaken by the survey company. Weighting is by age, gender, social class, region and level of education. Targets for the weighted data are derived from four sources:

1. The census
2. Large scale random probability surveys, such as the Labour Force Survey, The National Readership survey and the British Election Study
3. The results of the 2017 general election and 2016 referendum.
4. Official ONS population estimates

YouGov also use Active Sampling to ensure that the right people are invited in the right proportions. In combination with the statistical weighting, this ensures that results are representative of the country as a whole. Please see the following reference for more detail: YouGov. Panel Methodology. <https://yougov.co.uk/about/panel-methodology/> (Ref 22 in the manuscript).

“Weightings were calculated by YouGov, with the final data weighted to match the national profile of all adults aged 18 and over and applied during analyses to render the sample representative of UK adults.[22]” (Page 7, lines 239-241)

5. To be able to say something about a real change in the prevalence of visiting green space the authors should have used the pairedness of the data. If they compare question 1 and 2 (visit green space before and during the lockdown) they could have made specific combinations (yes/yes, yes/no, no/yes, no/no). Especially the yes/no group is interesting to see if there is exacerbated inequality (e.g. are there more C2DE subjects in this group compared to the yes/yes and also is this different from the ABC1 group). But also the no/yes group could be interesting to investigate further.

Response: We agree that it would be interesting to investigate the no/yes group further and conduct more detailed analysis into the yes/no group. Unfortunately, the number of those who reported not having visited greenspace before lockdown but visiting following restrictions (no/yes) was too small for such analysis (N=12, 1.1%).

Despite this limitation, the pairedness was taken advantage of as much as possible; we have now only analysed the data on visits since restrictions on movement were enforced for those who reported visiting green space before the restrictions.

We have included this point as a limitation of the study and recommend future research ensures a large enough sample of those who change patterns of behaviour:

“It would also be interesting to explore the change in patterns of use before and following movement restrictions for those that did not visit before the movement restrictions but did visit green space following restrictions.” (Page 17, lines 544-547)

6. Instead of doing binary logistic regression on decrease and increase separately the authors could also have done multinomial logistic regression in which they used the 3 categories (increased, the same, decreased) as their outcome (using 'the same' as reference category). This also applies to the agree/disagree variables.

Response: A reason that binary logistic regression was chosen was that the results are easier to interpret for some non-academic audiences, such as policy makers, stakeholders, and government officials, however you are correct that analysis on the 3-level outcomes could be undertaken. We are happy to report that we have replaced the original analyses with multinomial regression where appropriate. We also checked that ordinal logistic regression would not be better for the order implied in decrease, same, increase. The new models are applied to change in time visiting green space following restrictions on movement (increase,

same, decreased), and the three experience outcomes – mental health benefits, missed social interaction, and increased physical activity (agree, neither, disagree).

We present the results of the multinomial logistic regression models using average predicted probabilities for each group level. This is because predicted probabilities are easier to interpret than relative risks for non-academic audiences. Tables 2, 3 and 4 have been updated to present these new results. There were some very minor changes in the results when applying the new models with, for example, associations between change in time spent in green space and dog ownership, and reporting mental health benefits and dog ownership changing statistical significance. There were few substantive changes to the key findings, with only the identification of interactions between social grade and age for all 3 experience outcomes being noteworthy. These have been outlined in the results section:

“For the interactions associated with the experience outcomes, younger respondents from the higher social grade group had the highest probability of agreeing both that green space benefitted their mental health more, and that they missed social interaction in green space, following the movement restrictions. In contrast, younger respondents in the lower social grade group had the highest probability of disagreeing that mental health benefitted their mental health more. Older respondents in the lower social grade group had a higher probability of agreeing that they missed social interaction in green space than younger respondents. Finally, younger respondents had the highest probability of agreeing that they had increased physical activity following the movement restrictions compared to the older respondents in both social grade groups (Supplementary Table 5).” (Page9/10, lines 340-349)

7. In the supplementary tables the numbers do not always add up to the total. For example in S1 1659 white british and 86 any other white background makes 1659+86=1745 and not 1747 as stated in the table. Also in table S2 158 did not visit green space before restriction of which 119 did not own a dog and 40 owned a dog, but 119+40=159 and not 158. I did not check everything but I urge the authors to check all the numbers.

Response: Thank you very much for bringing this to our attention. An error was made in S1 that has now been amended. We can confirm that we have checked through these numbers and re-run the models to ensure that these are all correct. The individual counts and percentages for each demographic variable differ between the unweighted and weighted data, we have presented these results unweighted in the supplementary materials.

Reviewer: 2

Comments for the authors

The authors present the results of a cross-sectional survey in the UK exploring the effects of COVID-19 lockdowns on the visitation to and experience of green spaces. They find an overall decline in self-reported visitation to green spaces with demographic differences that indicate a possible exacerbation of inequality in use of green space. The manuscript is well written and will be of interest to readers of BMJ Open. The findings and conclusions drawn are supported by the methods and results presented. I have only minor comments below.

P4L35: I find myself wanting to know more detail about the social distancing measures here. It is important for an international audience who might have questions about what constituted “restrictions on movement” in the UK. For instance: were there any curfews in place? What was the scope of “exercise” – i.e. did it include walking pets?

Response: Thank you for highlighting this, we have now added more detail on what constituted the restrictions on movement in the UK at this time:

“These included only leaving home for limited purposes, such as medical needs, shopping for basic necessities (food and medicine), and exercising once a day alone/with members of your household.[23]” (Page 5, lines 169-171)

P5L41: The wording of the question in the survey does not necessarily imply frequency of visitation. The words “...time that you have spent visiting green spaces...” may be interpreted as number of visits or as time spent during each visit. In this way, two respondents may have answered “Increased” and one may have increased the frequency of green space visits while the other visited the same number of times as before the ‘lockdown’ but spent a longer duration of time in the green space. I would argue that the use of “frequency” in the title and throughout the manuscript is a little misleading given that the survey was not this specific.

Response: We agree that the wording of the question related to “time that you have spent visiting green spaces” may be unclear in regard to frequency. We have amended all wording related to frequency of visitation to time spent visiting green space throughout the manuscript to ensure that the results are not misleading. We have also updated the title of the manuscript.

Updated title: “Change in time spent visiting and experiences of green space following restrictions on movement during the COVID-19 pandemic: A nationally representative cross-sectional study of UK adults.” (Page 1, lines 1-3)

P6L29: It is not clear how what the significance of the dog ownership classification is. The results are reported but not discussed in the Discussion section. What are the implications of the results related to dog ownership?

Response: Thank you for highlighting this point, we have now provided the following text related to dog ownership within the discussion section:

“The results for dog owners suggest that the movement restrictions have had an overall negative impact on their experiences within green space. Although dog owners were more likely to have visited green space following the movement restrictions than those that did not own a dog, they were less likely to agree that they had increased physical activity and more likely to have decreased visitations following the restrictions. These results differ from research undertaken in Canada that reported findings that dog ownership was associated with more outdoor play and less indoor play in Canadian youth at the start of the pandemic.[48] However, the focus on youth rather than adults may explain this difference in findings. Instead, the difference found in visits after restrictions were enforced may be due to dog owners having to walk their dog/s in green spaces despite the pandemic restrictions. In comparison between our results and those of other studies, we are mindful of the difference between number of visits and time spent in green spaces as measures of ‘use’. It would be entirely possible to increase one at the expense of the other and our data were not well suited to unpacking the relationship between them.” (Page 16/17, lines 498-511)

P10L35: Discussion: this section is well written, however lacks reference to broader literature on the topic which decreases the relevance for an international audience. I advise including similar literature from other countries in Europe and further afield.

Response: We felt it important to initially compare results with studies from the UK for direct comparison with green space use and experiences before the pandemic/movement restrictions. To increase the relevance for an international audience we have added the following references and text:

33. Curtis DS, Rigolon A, Schmalz DL, et al. Getting out while staying in: Park use decreased during the COVID-19 pandemic, especially where park availability was low. Cent Open Sci OSF Prepr Published Online First: 2020. doi:10.31235/osf.io/9xzgf (Page 15, line 422):

“A similar study focused on change in time spent visiting parks using the Google COVID-19 Community Mobility Reports covering 620 counties across the United States(USA). They found a lower percentage decrease in park visits compared to our findings, reporting a 17-35% decrease in visits between 15th March-9th May 2020.[33] This difference may be explained by the focus on parks alone rather than different types of green space.”

34. Barton D, Haase D, Mascarenhas A, et al. Enabling Access to Greenspace During the Covid-19 Pandemic-Perspectives from Five Cities. Nat. Cities. 2020.https://www.thenatureofcities.com/2020/05/04/enabling-access-to-greenspace-during-the-covid-19-pandemic-perspectives-from-five-cities/ (accessed 23 Oct 2020) (Page 15, line 426):

“However, additional research exploring the Google COVID-19 Community Mobility Reports found that from 16th February-29th March 2020 park use decreased by 90% in Catalonia, 7% in Oslo, and 79% in New York County. In Stockholm, park use increased by 24% in the same timeframe.[34]”

46. Honey-Rosés J, Anguelovski I, Chireh VK, et al. The impact of COVID-19 on public space: an early review of the emerging questions – design, perceptions and inequities. *Cities Heal* 2020;:1–17. doi:10.1080/23748834.2020.1780074 (Page 16, line 488):

“Similarly, existing international research on public space use during COVID-19 restrictions worldwide state that skilled workers in the knowledge economy have shifted easily into online work from home, and can therefore make more use of green spaces during the movement restrictions.[46]”

48. Moore SA, Faulkner G, Rhodes RE, et al. Impact of the COVID-19 virus outbreak on movement and play behaviours of Canadian children and youth: a national survey. *Int J Behav Nutr Phys Act* 2020;17. doi:10.1186/s12966-020-00987-8 (Page 16, line 504):

“These results differ from research undertaken in Canada that reported findings that dog ownership was associated with more outdoor play and less indoor play in Canadian youth at the start of the pandemic.[48] However, the focus on youth rather than adults may explain this difference in findings. Instead, the difference found in visits after restrictions were enforced may be due to dog owners having to walk their dog/s in green spaces despite the pandemic restrictions.”

P17 Figure 1: Please remind the reader what “ABC1” and “C2DE” mean here.

Response: Thank you, this figure has now been changed.

VERSION 2 – REVIEW

REVIEWER	Judith Vonk University Medical Center Groningen, The Netherlands
REVIEW RETURNED	16-Nov-2020
GENERAL COMMENTS	The authors have answered most of my comments. I like it that they now did the analysis on green space visits after the lockdown only on those who visited the green spaces also before the lockdown (so comparing yes/yes to yes/no). I also like the multinomial logistic regression model. However, I do not like the use of predicted probabilities. Why not just present the ORs for the different outcomes. They state that PPs are easier to understand but actually I do not understand at all what is presented now. Also in the table you present p-values and you make some PPs bold to indicate that they are significant but what do you mean by that (what is tested here) and what does the given p-value indicate?

	A second comment that I still do not think that the interaction analyses make the paper stronger. The authors state that based on literature they expected some interactions. I would like to have more info on what they expected and if their results were in line with these expectations. Further, they present a figure with predicted probabilities. If they really want to present something like this than it would be much clearer to present just the prevalence instead of the predicted probabilities. They also state that these results presented in the figure should be interpreted with caution given the small numbers in the groups. Finally, they added a lot of text in the results describing the significant interactions. I regret to say that this is very difficult to follow and not really interesting. I think the paper is much stronger if the interaction analyses are not presented in the paper.
REVIEWER	Zander Venter Norwegian Institute for Nature Research, Norway
REVIEW RETURNED	16-Nov-2020
GENERAL COMMENTS	The authors have thoroughly addressed my comments. The revised manuscript reads much better. Most importantly, the conclusions are supported by the documented methods and results and the relevant shortcomings and limitations are presented well. I recommend the article for publication without further changes.

VERSION 2 – AUTHOR RESPONSE

Reviewer: 1

Comments to the author

The authors have answered most of my comments. I like it that they now did the analysis on green space visits after the lockdown only on those who visited the green spaces also before the lockdown (so comparing yes/yes to yes/no). I also like the multinomial logistic regression model.

Response: Thank you, we are pleased that you like the new analyses.

However, I do not like the use of predicted probabilities. Why not just present the ORs for the different outcomes. They state that PPs are easier to understand but actually I do not understand at all what is presented now.

Response: Thank you for these comments and suggestions. We acknowledge that our explanation and justification of predicted probabilities (PPs) was not very clear and have now added a guide to interpretation and a better explanation to the text (Page 6, lines 206-215). We

also see that some readers might prefer ORs or Risk Ratios and have therefore provided those in the supplementary material. We prefer to keep PPs in the main text. ORs only provides indications of the differences in likelihood between the base and other categories. Using the predicted probabilities gives a more detailed indication of the association between explanatory variables and the outcomes, as it clarifies what outcome each level of a factor is likely to give. Crucially, comparison between variables is also easier and, arguably, more valid. We hope that by providing both versions of the results, we can keep everyone happy! We have also amended the results section “Change in time spent visiting green space” to ensure that the predicted probability results are recorded more clearly (Page 8, lines 274-285).

Also in the table you present p-values and you make some PPs bold to indicate that they are significant but what do you mean by that (what is tested here) and what does the given p-value indicate?

Response: Apologies, this was a carryover error from the binary approach, and should have been removed. We have now removed the bold PPs and updated the table captions to ensure that the tables and p-values are clear (Table 3 and 4; pages 11 and 12). We calculated the p-values using likelihood-ratio tests, used to calculate the significance of the predictors to the model. For example, the results in table 3 for sex and change in time spent visiting green space shows that sex is a significant predictor to the model.

A second comment that I still do not think that the interaction analyses make the paper stronger. The authors state that based on literature they expected some interactions. I would like to have more info on what they expected and if their results were in line with these expectations. Further, they present a figure with predicted probabilities. If they really want to present something like this than it would be much clearer to present just the prevalence instead of the predicted probabilities. They also state that these results presented in the figure should be interpreted with caution given the small numbers in the groups. Finally, they added a lot of text in the results describing the significant interactions. I regret to say that this is very difficult to follow and not really interesting. I think the paper is much stronger if the interaction analyses are not presented in the paper.

Response: Thank you for your comments regarding the interaction analyses. We have removed the interaction analysis, including the figure, and agree that this makes the paper a more coherent read.

Reviewer: 2

Comments to the author

The authors have thoroughly addressed my comments. The revised manuscript reads much better. Most importantly, the conclusions are supported by the documented methods and results and the relevant shortcomings and limitations are presented well. I recommend the article for publication without further changes.

Response: Thank you very much for your comments, we are glad that you think that the manuscript reads much better and that the conclusions are well supported.